# Allergic Bronchopulmonary Aspergillosis (ABPA) in the Era of Cystic Fibrosis Transmembrane Conductance Regulator (CFTR) Modulators

**DOI:** 10.3390/jof10090656

**Published:** 2024-09-18

**Authors:** Paulami Chatterjee, Carson Tyler Moss, Sarah Omar, Ekroop Dhillon, Carlos Daniel Hernandez Borges, Alan C. Tang, David A. Stevens, Joe L. Hsu

**Affiliations:** 1Division of Pulmonary, Allergy and Critical Care Medicine, Stanford University School of Medicine, Stanford, CA 94305, USA; paulami@stanford.edu (P.C.); saromar@stanford.edu (S.O.); edhillon@stanford.edu (E.D.); 2Department of Medicine, Stanford University School of Medicine, Stanford, CA 94304, USA; 3Department of Internal Medicine, Alta Bates Summit Medical Center, Sutter Health, Oakland, CA 94609, USA; hernc32@sutterhealth.org; 4Department of Medicine, Keck School of Medicine, Los Angeles, CA 90089, USA; actang@usc.edu; 5Division of Infectious Diseases and Geographic Medicine, Stanford University Medical School, Stanford, CA 94305, USA; stevens@stanford.edu

**Keywords:** allergic bronchopulmonary aspergillosis, cystic fibrosis, *Aspergillus fumigatus*, *Aspergillus* proteases, allergic inflammation, asthma, Th2 immune pathway, CFTR modulators, elexacaftor–tezacaftor–ivacaftor (ETI)

## Abstract

Allergic bronchopulmonary aspergillosis (ABPA) is a hypersensitivity disease caused by *Aspergillus fumigatus* (*Af*), prevalent in persons with cystic fibrosis (CF) or asthma. In ABPA, *Af* proteases drive a T-helper cell-2 (Th2)-mediated allergic immune response leading to inflammation that contributes to permanent lung damage. Corticosteroids and antifungals are the mainstays of therapies for ABPA. However, their long-term use has negative sequelae. The treatment of patients with CF (pwCF) has been revolutionized by the efficacy of cystic fibrosis transmembrane conductance regulator (CFTR) modulator therapy. Pharmacological improvement in CFTR function with highly effective elexacaftor/tezacaftor/ivacaftor (ETI) provides unprecedented improvements in lung function and other clinical outcomes of pwCF. The mechanism behind the improvement in patient outcomes is a continued topic of investigation as our understanding of the role of CFTR function evolves. As ETI therapy gains traction in CF management, understanding its potential impact on ABPA, especially on the allergic immune response pathways and *Af* infection becomes increasingly crucial for optimizing patient outcomes. This literature review aims to examine the extent of these findings and expand our understanding of the already published research focusing on the intersection between ABPA therapeutic approaches in CF and the rapid impact of the evolving CFTR modulator landscape. While our literature search yielded limited reports specifically focusing on the role of CFTR modulator therapy on CF-ABPA, findings from epidemiologic and retrospective studies suggest the potential for CFTR modulator therapies to positively influence pulmonary outcomes by addressing the underlying pathophysiology of CF-ABPA, especially by decreasing inflammatory response and *Af* colonization. Thus, this review highlights the promising scope of CFTR modulator therapy in decreasing the overall prevalence and incidence of CF-ABPA.

## 1. Background

### 1.1. Cystic Fibrosis

Cystic fibrosis (CF) is a multisystemic, autosomal recessive genetic disorder stemming from the dysfunction of the cystic fibrosis transmembrane conductance regulator (CFTR) gene that encodes for a transmembrane protein essential to the transport of chloride and bicarbonate [1,2]. Although clinical evidence of CF has been reported since the 17th century, the disease was only recognized as a distinct clinical entity by Dorothy H. Andersen in 1938 [1,3,4]. According to a recent study, worldwide there are an estimated 162,428 people with CF [5], with a higher prevalence among people of European ancestry, especially those of Northern European descent. The incidence is estimated to be 1 case per 3000 live births [6]. Notably, approximately 1 in 25–30 individuals of European ancestry are carriers of a pathogenic mutation of the CFTR gene [7]; however, CF is also present in non-European populations [8]. Mutations in CFTR drive the pathogenesis of CF and can be divided into six different classes (class I–VI) according to their functional deficits [9,10]. The discovery and characterization of the different CFTR mutations heralded a new era of targeted therapies for CF that includes the promising advent of CFTR modulator therapy that directly augments deficiencies related to CFTR mutations [11,12,13].

Until the advent of CFTR modulators, treatment was limited to the sequelae of CF with a focus on improving nutritional status with enzyme replacement, airway clearance with mucolytics, and treating recurrent respiratory infections [14]. While these interventions remain a cornerstone of treatment, they do not address the underlying cause of the disorder. The first CFTR modulator, ivacaftor, was approved by the United States Food and Drug Administration (FDA) in 2012 [15]. Although ivacaftor was a breakthrough in CF therapy, it did not cover all CFTR mutations, including homozygous F508del, which has a median life expectancy of 37 years [16]. In 2015, the first combination modulator therapy entered the market, containing ivacaftor and lumacaftor. Further advances were made by adding a third modulator, elexacaftor, introducing the practice of using triple therapy with CFTR modulators. In 2023, the FDA expanded the indications for elexacaftor–tezacaftor–ivacaftor (ETI) to include pwCF above the age of 2 years old with at least one F508del mutation or a gene with proven response in vitro, expanding eligibility to 92% of pwCF [17].

The characteristic pathophysiology of CF is the production and accumulation of dense mucus [1,3,6]. In CF, abnormally viscous mucus is found in various affected organs including the pancreas, lungs, GI tract, and reproductive tract. This thick mucus was associated with complications such as pancreatic insufficiency, gallstones, meconium ileus, infertility, sinusitis, and bronchiectasis, an inflammation-induced architectural distortion of the airways [1,3]. Electrolyte abnormalities in the airways are another manifestation due to greater electronegative differences across the nasal epithelium resulting from impermeability to chloride ions [1]. Furthermore, dysfunctional CFTR protein causes enhanced epithelial sodium channel (ENaC) activity leading to increased sodium absorption from the lumen [1,18]. These electrolyte disturbances result in the depletion of the volume of the airway surface liquid (ASL), producing a thick and viscous mucus that impairs mucociliary clearance, predisposing patients to increased inflammation, recurrent infections, and eventually, bronchiectasis.

### 1.2. Allergic Bronchopulmonary Aspergillosis

Allergic Bronchopulmonary Aspergillosis (ABPA) is a pulmonary hypersensitivity disease mediated by an allergic response instigated by inhaled conidia of *Aspergillus fumigatus* (*Af*) and occasionally of other *Aspergillus* species, commonly found in natural environments, including water, soil, decaying organic matter, and indoor environments [19,20,21]. ABPA involves complex immune pathways, and the factors underlying pathophysiology are not yet fully understood (Figure 1). The predisposing factors include atopy, immunogenetic HLA-restricted phenotypes, mutations in the CFTR gene, polymorphisms of the collagen region of the surfactant protein A2 and other collectins such as mannose-binding lectin, physicochemical characteristics of respiratory secretions, and environmental exposure history [22,23,24,25]. *Af* colonization of the respiratory tract is another factor associated with the pathophysiology of this disease [26]. The small size of *Af* conidia (3–5 µm) allows the inhaled conidia to bypass the mucociliary clearance and adhere to bronchial and type II alveolar epithelial cells [27]. There the conidia can persist and germinate within the respiratory system, leading to the growth of hyphae, resulting in mucus plugs seen in the expectorated sputum of ABPA patients [22]. Ordinary inhalation of this fungi does not typically trigger adverse reactions, but in individuals suffering from pulmonary dysfunction (e.g., CF, asthma, and chronic obstructive pulmonary disease [COPD]) or immunosuppressive conditions negative outcomes can emerge [28,29].

Atopy is a risk factor for hypersensitivity reactions, with ABPA occurring in 22% of patients with atopy and only 2% in patients without atopy [24,30]. It is well established that CFTR dysfunction results in impaired mucus production and clearance, thus promoting microbial colonization in the airway [31,32]. Patients with CF are at a baseline higher risk for ABPA, because up to 60% of CF patients are colonized with *Af* [33]. These fungi can function as allergens, especially in the presence of a Th2-activated state. Increased frequency and/or activity of these *Af*-specific Th2 CD4+ cells are established as a major contributor to ABPA in CF patients [24]. Studies have reported that CFTR-deficient animals develop an excessive Th2-biased inflammatory response to *Af* exposure that directly contributes to airway inflammation and lung damage [34,35]. Thus, in CF, CFTR-dependent altered regulation of T-cell cytokine secretion leads to a shift toward a pro-inflammatory state and a predominant Th2 response with exaggerated immunoglobulin E (IgE) levels, peripheral eosinophilia, and pulmonary infiltrates, a predisposition for ABPA [31,34,36,37,38,39,40,41,42,43]. Consequently, this persistent colonizing *Af*-mediated IgE response leads to a series of complications and acute worsening of the respiratory status including recurrent inflammation of the lung, *Aspergillus* bronchitis, bronchiectasis, and pulmonary fibrosis [21,30]. This defective Th2 response also drives the development of multispecies cross-reactive Th2 cells as a potential risk factor for ABPA [44].

Among those with hypersensitivity immune conditions, colonization of the airways by *Af* can lead to heightened immune responses, contributing to the development of ABPA. *Af* secretes allergens with proteolytic activity (proteases), which are capable of further compromising mucociliary clearance, breaching the airway epithelial barrier, and activating the innate immune system of the lung, including the production of several cytokines by epithelial cells [22,45]. These fungal proteases have been consistently implicated in the initiation and hyperresponsiveness of allergic response in ABPA and other allergic lung diseases including severe asthma with fungal sensitization (SAFS) [26,46,47,48,49,50,51]. The immune system response to *Af* conidia and proteases begins with the recognition of the pathogen-associated molecular patterns (PAMPs) by innate immune cells [27]. Innate immune cells recognize PAMPs through pattern recognition receptors (PRRs) present in epithelial cells and antigen-presenting cells (APCs) such as dendritic cells [27,43,52,53]. These PRRs recruit T-helper (Th) cell pathways [52]. Th1 is shown to be prevalent in eliminating fungal infection in healthy hosts, which is associated with increased levels of interleukin (IL)-2, IL-12, and interferon-gamma (IFN-γ). Th2 is the main pathway activated in patients diagnosed with ABPA [19,26,27,43,52]. However, unlike Th1 activation, the Th2 response does not eliminate *Af* [54]. Th2 inflammation represents the acute, but persisting, inflammatory airway response associated with CXCR4+ granulocytes, with the consequent production of numerous cytokines (including the alarmins, thymic stromal lymphopoietin (TSLP), interleukin (IL)-25, and IL-33) and chemokine (CCL17) that are associated with promoting the Th2 inflammatory response [27,54,55]. These mediators both directly drive the generation of the classic Th2 cytokines IL-4, IL-5, IL-9, and IL-13 from type 2 innate lymphoid cells (ILC2s), and mast cells but are also responsible for the differentiation of the antigen-specific Th2 effector cells that also produce these cytokines [28,31,55,56,57]. Loss of epithelial barriers allows for allergens and microbes to access stromal tissue and further promotes the activation of Th2 effector lymphocytes and B cells [31]. Recent in vitro and in vivo studies have shown Th2-promoting cytokines (IL-25, IL-33, and TSLP) [58,59] and Th2 cytokines (IL-4, IL-5, and IL-13) [35] to be upregulated by *Af* conidia. IL-4, IL-5, and IL-13 have all been shown to activate pathways relating to the production of IgE and IgG4, eosinophil activation, and mast cell degranulation [52,54,55,60,61]. Thus, this heightened Th2 response results in a detrimental effect on the patient resulting in shortness of breath, eosinophilia, mucus production, bronchiectasis, airway inflammation, and the poor asthma control that characterizes ABPA [62,63]. The increased immune response and chronic inflammation in ABPA worsen bronchiectasis, causing frequent respiratory infections and a progressive reduction in lung function [64].

### 1.3. CF-ABPA

As CF is a major precursor to ABPA, it is important to understand the relationship between CF and ABPA (Figure 2). ABPA commonly affects individuals with asthma (prevalence 1–2%, although rates of up to 13% have been reported) and CF (prevalence 5–15%, although rates of up to 25% have been reported), with variation among geographical regions, patient cohorts, age, and diagnostic criteria [21,22,23,24,29,65,66,67,68,69,70,71]. The prevalence of ABPA is increased in patients with CF who are male, are adolescents, have lower lung function, have a history of wheezing or asthma, or have *Pseudomonas aeruginosa* in the sputum [24]. Notably, adult CF patients tend to have a higher prevalence compared to children [72]. Regional disparities are also evident, with the United States and Canada estimating a lower ABPA prevalence of 2% [65] compared to a higher 7.8% prevalence in Europe [73,74]. Furthermore, diagnostic criteria greatly influence prevalence estimates, as demonstrated by differing rates within the same region [67]. A systematic review by Maturu and Agarwal in 2015 highlighted a substantial publication bias and heterogeneity among studies, with ABPA prevalence ranging from 3% to 25%, resulting in a pooled prevalence of 8.9% (95% confidence interval (CI): 7.4–10.7%) [72]. Ethnic and racial disparities in ABPA prevalence are evident beyond Western countries such as the United States and Europe, extending to Asian countries, particularly India. In India, the ABPA prevalence is notably higher, primarily driven by the increased incidence of respiratory conditions. ABPA prevalence among asthmatic patients in India is as high as 13% [62,75], matching the rates observed in Western CF patients, despite differences in population demographics and health conditions.

According to the 2014 CF Foundation (CFF) Patient Registry Annual Data Report, pwCF had a 5% prevalence of ABPA across all age groups, with 3.1% in those under 18, and 7% in those aged 18 years and over [76]. Interestingly, a modest but promising improvement in these numbers was recorded in the 2022 CFF Patient Registry Annual Data Report, where pwCF had a 4.3% prevalence of ABPA across all age groups, with 1.4% in individuals under 18 and 6.5% in those aged 18 years and over [77]. This improvement in the prevalence of CF-ABPA across different age groups could be attributed to the temporal incorporation and widespread application of highly effective CFTR modulator therapies during this time. As CFTR therapy becomes increasingly prevalent in CF management, understanding its potential impact on ABPA, particularly on the allergic immune response pathways and *Af* infection becomes increasingly important for optimizing patient outcomes. While data specifically focusing on CF-ABPA patients are limited, findings from epidemiologic and retrospective studies suggest the potential for CFTR modulator therapies to positively influence pulmonary outcomes in this population as well [12,31,32,78,79,80,81,82,83,84,85,86,87,88,89,90,91,92,93,94,95,96,97,98,99,100,101,102,103,104,105,106,107].

Due to the nascency of CFTR modulator therapy, much remains unknown about the long-term disease-modifying effect. There are currently limited case reports documenting ABPA in patients receiving CFTR modulator therapy. To further expand our knowledge about ABPA cases during CFTR modulator therapy, we performed a thorough literature search in the PubMed online library with search terms “eosinophilia”, “fungus”, “allergic”, “CFTR modulator”, “aspergillosis”, “*Aspergillus*”, and “ABPA” including CFTR clinical trials. In total, we retrieved 30 studies (Appendix A) [78,79,80,81,82,83,84,85,86,87,88,89,90,91,92,93,94,95,96,97,98,99,100,101,102,103,104,105,106]. One ABPA-associated term was found in the Taylor-Cousar et al., 2017, study of tezacaftor/ivacaftor where a placebo patient with ABPA was listed as an adverse effect [101]. This search yielded two case reports [41,107] of patients experiencing symptomatic ABPA during CFTR modulator therapy, which we discuss in detail in the subsequent Section for ‘Treatment of ABPA’. Theoretically, by regaining lung function and addressing the pathophysiology of CF, there is the promise that these medications could decrease the prevalence and incidence of CF-ABPA.

## 2. Diagnosis of ABPA

ABPA was first diagnosed and reviewed by Hinson et al., in 1952, who discussed the *Aspergillus* mycology, manifestation, differential diagnosis, and treatment in eight case reports [108]. Contemporary data from the CFF 2022 annual registry report shows that 4.3% of pwCF experienced ABPA as a complication of their underlying disease [77]; however, this figure is likely an underestimate, given the overlap in manifestations of ABPA and CF. Additionally, the similarities of ABPA to other pathologies (i.e., chronic eosinophilic pneumonia, Hyper-IgE Syndrome, persistent asthma with lobar collapse) have made diagnosis more difficult and impaired the establishment of standardized diagnostic criteria [67,109,110]. Multiple diagnostic criteria have been proposed for ABPA since it was first described in 1952. Common factors of these various criteria include *Af*-specific immunoglobulin, asthma, elevated total serum IgE levels (greater than 500 IU/mL), peripheral blood eosinophilia, cutaneous hypersensitivity reaction to *Af*, central bronchiectasis on chest CT, presence of pulmonary opacities on chest radiograph radiography and elevated serum IgE and/or IgG to *Af* [24,63,65,66,74,111,112,113,114,115,116]. Although typically not used as diagnostic criteria, pulmonary function tests (PFTs) can identify acute or subacute clinical deterioration that may be associated with ABPA. The most recent and widely used diagnostic standard is the International Society for Human and Animal Mycology (ISHAM) ABPA Working Group (ISHAM-AWG) criteria, which was first proposed in 2013 and updated in 2022 [19,110,117]. The ISHAM-AWG criteria have seven possible criteria (Table 1) using only serology and imaging. If a patient satisfies any of these variations, they may be considered to have ABPA. 

In 1982, Patterson et al. were the first to propose five clinical stages of ABPA [113]. Although the clinical course of ABPA varies among patients, these five recognized stages are useful for following the status and trajectory of a given patient [30,113,118]. It is noteworthy to mention that ABPA does not necessarily sequentially evolve through these stages. At presentation, it is not always clear who will enter remission, who will have recurrent disease, or who will progress. Regardless, early diagnosis and treatment are thought to be associated with a lower risk of advanced disease in the future [30,118]. A more recent clinical staging of ABPA has been published by Agarwal et al., which is a derivative of stages proposed by Patterson and takes into account the most recent ISHAM working group diagnostic criteria (Table 1) [19,110].

ABPA is challenging to diagnose and remains underdiagnosed in many pwCF as the clinical features of this condition overlap significantly with that of CF [24,74]. Moreover, Stevens et al. reported that there is no uniformity in making the diagnosis of CF-ABPA in the literature despite being highly prevalent in CF [24]. As per the CFF consensus criteria, minimal diagnostic criteria for ABPA in CF patients require evidence of acute or subacute clinical deterioration not attributable to another etiology, total serum IgE level > 1200 ng/mL while off steroids, evidence of *Af* sensitization and of the following: (1) serum precipitins to *Af*; (2) demonstration of *Af*-specific IgG antibodies in vitro; or (3) new or recent abnormalities on pulmonary radiologic imaging, which do not respond to antibiotics and chest physiotherapy [24,74]. Stevens et al. discussed in detail the characteristic imaging findings (bronchiectasis, pulmonary infiltrates, mucus plugging, pleural thickening) seen in patients with CF-ABPA and compared with their appearance in patients with CF without ABPA. The authors also pointed out that large-scale epidemiological studies are dependent, in their estimates of prevalence, on whether CF centers systematically screen for ABPA and have the appropriate clinical and laboratory facilities for testing [24].

While ABPA is seen almost exclusively in patients with asthma or CF, there have been a few reported cases in the absence of these diseases. Case reports of ABPA diagnoses have included patients with atopy due to allergic rhinitis and raised total IgE [119], mycoplasma pneumonia [120], allergic fungal sinusitis, bronchocentric granulomatosis, hyper-IgE syndrome, and chronic granulomatous disease [26].

## 3. Treatments for CF-ABPA

Therapies for ABPA are multifactorial, targeting both the inflammation pathognomonic of the disease, and reducing infection and colonization from *Aspergillus* which drives the production of antigens (Table 2). 

### 3.1. Corticosteroids

Synthetic corticosteroids such as prednisone and prednisolone are well-known for their capacity to reduce lung inflammation and mitigate severe lung damage [121,122]. Studies have demonstrated that corticosteroid therapy can lead to significant improvements in PFT metrics such as forced expiratory volume in one second (FEV_1_) and forced vital capacity (FVC) [121,122]. Despite this benefit, prolonged use may cause deleterious adverse effects as described in Table 2 [20,121,122]. Greater than 90% of patients taking corticosteroids for longer than 60 days experience a medication-associated adverse effect, including those taking low-dose prednisone (≤7.5 mg daily) [123]. Most concerning is that long-term use of high-dose corticosteroids is a significant risk factor in the development of more serious *Af* infections, particularly invasive aspergillosis (IA) [124]. Nonetheless, the judicious use of corticosteroids has been a cornerstone in the management of CF-associated ABPA for the past 45 years [111].

### 3.2. Antifungal Agents

While corticosteroid use can decrease inflammation from *Af*, antifungal therapy provides an opportunity to reduce the fungal burden that triggers the inflammatory response. The Infectious Diseases Society of America (IDSA) recommends that pwCF with frequent exacerbations or worsening lung function be treated with itraconazole to minimize corticosteroid use [125]. Stevens et al., in a seminal randomized controlled trial, were the first to demonstrate the efficacy of itraconazole as an effective adjuvant therapy in corticosteroid-dependent ABPA, resulting in an improvement of symptoms and decreased corticosteroid dose without additional drug toxicity [126]. Later studies concluded that prednisolone and itraconazole have similar effects on time to exacerbation; however, itraconazole resulted in fewer side effects compared to the corticosteroid. Additional triazoles such as voriconazole, isavuconazole, and posaconazole have demonstrated efficacy in ABPA treatment and are applied to clear *Aspergillus* infections and colonization, thereby constraining the inflammatory responses related to ABPA [127,128,129]. 

Triazole therapy for ABPA is associated with improvements in FEV_1_ and FVC measurements; however, data specific to CF-ABPA is limited [127,128,129]. A case report from Mainz et al. 2021 describes a 13-year-old female with compound heterozygous CF who, despite undergoing improvement in lung function following initiation of ivacaftor monotherapy, experienced a severe ABPA exacerbation. Treatment with systemic corticosteroids and itraconazole resolved the flare and returned the patient to her baseline; however, this case highlights continued susceptibility to ABPA, even with the recovery of CFTR function following modulator therapy [107]. 

Although triazoles are typically well tolerated, a diverse array of adverse effects has been reported by studies [130,131], which are described in Table 2. Through inhibition of CYP3A4, azoles are prone to numerous interactions with drugs that are cleared by the liver, though this is most pronounced with ketoconazole, which is not used in the treatment of IA or ABPA. Drug levels of cyclosporine, tacrolimus, and methylprednisolone may be elevated with concurrent triazole use, increasing the risk of drug toxicity. With the increasing prevalence of CFTR modulator use, attention must be paid to potential drug interactions with critical CF therapies.

### 3.3. Biologics

Given the poor side effect profile of prolonged corticosteroid use and concern for drug interactions with triazole antifungals, biologics provide an alluring alternative steroid-sparing regimen for CF-ABPA [132,133]. Biologics, specifically monoclonal antibodies (mAbs), are capable of impairing various pathways of inflammation implicated in the pathogenesis of ABPA. For example, omalizumab specifically targets the Fc region of IgE, which is a key component of hypersensitivity reactions [122,134,135,136]. By binding to the Fc region, omalizumab effectively blocks IgE from interacting with receptors on immune cells, thereby reducing inflammation and controlling symptoms associated with ABPA [135,137]. A randomized, cross-over trial of 13 patients by Voskamp et al. found that following 4 months of treatment with omalizumab 750 mg monthly, subjects experienced a significant decrease in exacerbations (2 vs. 12), basophil sensitivity to *Af*, and reduced fraction exhaled nitric oxide (FeNO) (30.5 vs. 17.1 ppb), a marker of lower airway inflammation [138].

There are multiple other studies describing the off-label use of omalizumab (anti-IgE) [138], mepolizumab (anti-IL5) [139], dupilumab (anti-IL4Rα) [140], and benralizumab (anti-IL5R) [141]. Given the role of IgE, IL-4, IL-5, and IL-13 in the pathogenesis of ABPA, there is an immunologic basis for their use. However, supporting evidence for their use in randomized control trials remains lacking [141]. To date, biologic therapy has been utilized primarily in patients with ABPA refractory to other treatments [135,137], such as a patient described by Boyle et al., who was treated with mepolizumab due to inability to tolerate high-dose prednisone, triazole, or omalizumab [41]. This patient, a 43-year-old female with severe, recurrent CF-ABPA, characterized by exacerbations with eosinophilia, elevated serum total IgE, anti-*Aspergillus* IgE, and recurrent lobar collapse, received high-dose prednisolone, azole, and omalizumab during prior exacerbations, though she experienced significant side effects to each, including oral candidiasis, visual disturbances, and severe arthralgias and myalgias. Given previous intolerance to these standard therapies, subsequent recurrence was treated with mepolizumab 100 mg, which was well tolerated and allowed for cessation of prednisolone, six weeks later. FEV_1_ stabilized at 45 percent predicted (prior baseline) and she remained free from exacerbations over the following 20 months. Later initiation of tezacaftor/ivacaftor therapy was associated with no significant symptomatic or spirometry improvement, though no adverse drug reactions were noted as well [41]. Biologics are typically well tolerated. The most significant adverse effect of mAbs is anaphylaxis; however, this is more common in chimeric antibiotics, and less so in humanized (omalizumab, mepolizumab, benralizumab) and human-derived (dupilumab) antibodies [142]. Other adverse effects are listed in Table 2.

Despite the promising outcomes, larger clinical trials specifically targeting CF-ABPA patients are warranted to further assess the efficacy and safety profile of mAbs in this population, especially regarding drug interactions with CFTR modulators. 

## 4. CFTR Modulator Therapy

While not a treatment for ABPA, CFTR modulator therapies have been proposed to have an impact on decreasing ABPA exacerbations due to their disease-modifying effect on other acquired CFTR dysfunction diseases such as COPD and asthma [143]. 

### 4.1. Types of CFTR Modulators

CFTR modulators are medications that directly augment deficiencies related to CFTR mutations (Table 3). Multiple classes of modulators exist; however, two are primarily utilized: potentiators and correctors [144]. Potentiators enhance the probability of the CFTR channel opening (gating), allowing chloride ions to flow through the channel, whereas correctors bind to misfolded CFTR proteins, assisting in the folding process, and ultimately increasing their stability and correctly folded conformation. Amplifiers are a newer class of modulators aimed at increasing protein production; phase II trials with the amplifier nesolicaftor have been completed with positive findings, though there remain no FDA-approved amplifiers [145]. Similarly, read-through agents, such as ataluren, have been proposed to negate premature stop signals during CFTR protein synthesis; however, studies assessing use in class I mutations have been insufficient to suggest their use [146]. CFTR stabilizers remain as a theoretical class of modulators seeking to increase the stability of CFTR protein on cell surfaces [147]. 

### 4.2. CFTR Modulator Therapies Clinical Evidence and Current Practice Trends

Ivacaftor a CFTR potentiator was first studied in G551D mutations, which impairs CFTR gating (class III). Clinical trials demonstrated improvement in predicted FEV_1_ by 10.6%, which was sustained through week 48; additionally, subjects experienced significant weight gain (2.7 kg), decreased patient-reported symptoms (measured via Cystic Fibrosis Questionnaire—Revised), and decreased incidence of pulmonary exacerbations by 55% [78]. Subsequent studies in subjects with non-G551D-CFTR mutations similarly displayed increases in lung function and BMI, and decreased pulmonary exacerbation and antibiotic use [148,149]. Lumacaftor, a corrector, was first studied in homozygous F508del bronchial epithelial cells where it improved CFTR processing and enhanced chloride secretion to 14% [150]. However, both lumacaftor and ivacaftor were shown to be ineffective in vivo for the treatment of homozygous F508del [151,152].

By combining correctors and potentiators, combination therapy can synergistically increase the amount of functioning CFTR protein present on membranes and the probability of gating, respectively [153]. When used in combination, lumacaftor–ivacaftor yielded modest improvement in pulmonary function, with a 2.6 to 4.0% increase in percent predicted FEV1 and decreased the rate of pulmonary exacerbations by 30 to 39% compared to placebo [90]. A similar combination of tezacaftor–ivacaftor proved efficacious, improving FEV_1_ by 4.0 percent predicted and decreasing exacerbations by 35% vs. placebo [101]. While both combination therapies yielded statistically significant improvements, these findings were not as profound as the initial ivacaftor-G551D trial.

The addition of elexacaftor, another corrector that also exhibits a potentiator effect [154], formed the triple-therapy combination ETI therapy. Studies have demonstrated notable improvements in FEV_1_ and FVC, along with reduced pulmonary exacerbations and improved quality of life [105,155,156]. Initial trials demonstrated robust PFT improvement in subjects carrying homozygous F508del mutations, including a 13.8% increase in percent predicted FEV_1_ and a 63% reduction in pulmonary exacerbation rate within 4 weeks of initiation [105]. Triple-therapy modulators provide equivalent improvement to patients with homozygous F508del as ivacaftor monotherapy does for those with G551D mutations. Outcomes of ETI remain consistent when compared to tezacaftor–ivacaftor, providing superior FEV1 improvement by 10% predicted, in addition to significant differences in weight gain and patient-reported symptoms [106]. An open-label post-marketing study found that these benefits were sustained over 144 weeks of prolonged use [157].

Models predicting the life expectancy of patients taking ETI estimate that there is a 33.5-year improvement when compared to supportive care, with the greatest impact in those ages 12–17 years old who are projected to experience an increase of 45.4 years [158]. The model also predicted the relative ratio of pulmonary exacerbation to be 0.22 compared to supportive care, which is consistent with studies corroborating decreased exacerbation events [159,160]. However, there is a paucity of evidence specifically describing the role of CFTR modulator therapy in CF-ABPA. Theoretically, by regaining lung function and addressing the pathophysiology of CF [12], there is hope that these medications decrease the prevalence and incidence of CF-ABPA. In what follows, we will outline the available evidence between CFTR modulator therapy and CF-ABPA. Despite the documented efficacy of current therapies, inherent limitations are present. Addressing treatment-related complications, potential drug interactions with CFTR modulators, and long-term management challenges are critical aspects that require attention [161].

## 5. CFTR Modulator Effects on CF Pathophysiology

In this Section, we have outlined the effect of CFTR modulators on CF pathophysiology, focusing on the clinical and physiologic outcomes of CF-ABPA.

### 5.1. CFTR Modulator Restores Epithelial Homeostasis in CF

Epithelial cells are self-organized, dynamic structures that serve as an interface between an external environment and an organ’s internal microenvironment [162]. Epithelial homeostasis ensures the proper functioning and health of our organs by maintaining a delicate equilibrium between cell turnover and protective functions. Previous studies reported a disruption of bronchial epithelial cell homeostasis in CF and an altered immune cell repertoire with mostly immature pro-inflammatory neutrophils in CF airways [163,164]. It has been shown that treatment with lumacaftor/ivacaftor enhances the repair of CF epithelial cell monolayer injury [165]. By single-cell RNA-sequencing of epithelial and immune cells obtained by nasal swabs from children with CF and healthy children, Loske et al. found that in CF, the proportion of CFTR+ cells is reduced in most epithelial cell types, but partially restored after initiation of ETI [166]. Analysis of the proportions of CFTR+ cells within nasal epithelial cells revealed reduced CFTR+ basal, club, and goblet cells in children with CF at baseline compared to healthy children [166]. ETI treatment increased the proportion of CFTR+ cells in club and ciliated cells to the level of healthy controls while the proportion of CFTR+ cells in goblet cells remained decreased compared to healthy controls [166]. In ionocytes, which consistently express the majority of CFTR in airway epithelial cells in adults [163,166,167,168], the proportions of CFTR+ cells did not change after initiation of ETI therapy [166]. A low number of CFTR+ immune cells, representing a frequency of 0–1.2% of the corresponding immune cell type, were also found to be unaffected by ETI [166]. CFTR regulates many mechanisms in epithelial physiology, such as maintaining epithelial surface hydration and regulating luminal pH, which is an important arbiter of epithelial barrier function and innate defense, particularly in the airways and GI tract [169]. Thus, by restoring CFTR+ epithelial cell type, CFTR modulators maintain epithelial equilibrium and restore defense against pathogens in airways. However, whether nasal/upper airway inflammation mimics the bronchial/lower airway inflammation in CF, and whether CFTR modulator therapy alters upper airway and lower airway inflammation in a similar manner, is yet to be determined [170].

### 5.2. CFTR Modulators Reduce Pro-Inflammatory Phenotypes in CF

Exposure of CFTR F508del homozygous epithelial cells in vitro to lumacaftor/ivacaftor markedly dampens the intracellular signaling cascade in response to inflammatory stimuli and transcription of IL-8 [171]. In animal models of CF, studied in pigs, rats, and ferrets, that express G551D-CFTR, CFTR genes are shown to be potentiated effectively by ivacaftor, where ivacaftor enhanced CFTR channels opening [170,172,173]. G551D-CFTR ferrets administered ivacaftor beginning in utero are spared the airway pathology seen in CF ferrets not receiving modulators, but then develop airway inflammation when ivacaftor therapy is withdrawn [173]. G551D-CFTR rats raised for 6 months in the absence of ivacaftor demonstrate increased markers of airway inflammation in bronchoalveolar lavage (BAL) fluid compared with wild-type rats [173]. Interestingly, some of the inflammatory markers (IFN-γ, IL-1α, and IL-1β) reduced significantly following 1 week of ivacaftor treatment, whereas the decrease in IL-6 and TNF-α concentrations was not statistically significant, indicating that restoration of CFTR activity in airways with established disease may not completely reverse inflammation [170,174]. Another study reported decreased levels of IL-1β and IL-6 in the upper airway lining fluid obtained from pwCF starting ivacaftor [175]. Furthermore, CFTR deficiency mediated impaired IFN signaling, and MHC gene expression was shown to be partially restored by ETI therapy [166]. This study also reported that genes upregulated by ETI were involved in antigen presentation and processing, whereas genes associated with IL-1 and TNF-α signaling pathways were among the genes repressed after initiation of ETI compared to baseline. Furthermore, ETI therapy reduced the activation of IL-1 downstream mediators, expression of pro-inflammatory chemokines like CCL3 and CCL4, and reduced the elevated transcription of AQP9 (encoding aquaporin 9), which is involved in IL-1β secretion, in monocyte-derived non-resident macrophages and neutrophils of children with CF [166]. Similar to IL-1 signaling, a general ETI-dependent decrease in expression of TNF-α to levels comparable to those of healthy children and a reduced transcriptional activation of the TNF signaling pathway was observed in macrophages and neutrophils [166]. TNF-α is one of the key drivers of neutrophilic inflammation, a hallmark of CF airway disease, and levels are reportedly elevated in vivo [176]. TNF-α along with another inflammatory cytokine IL-17 has been established to induce bicarbonate (HCO_3_^−^) secretion by upregulating pendrin, an apical Cl^−^/HCO_3_^−^ exchanger [176]. This study also reported that TNF-α along with IL-17 increased the efficacy of CFTR modulators and further alkalinized the ASL in CF, which helps in restoring respiratory host defense [176]. Thus, it indicates a positive correlation between airway inflammation and CFTR modulator-induced lung function improvements and further establishes that inflammation is a key regulator of HCO_3_^−^ secretion in CF airways and that it enhances the efficacy of CFTR modulators. 

### 5.3. Role of Modulators on Myeloid Homeostasis

Prior studies reported that restoration of CFTR activity reverses some CF immune cell defects in in vitro settings [170]. Defects in adhesion and trafficking observed in monocytes from persons with CF with F508del homozygous mutation could be reversed by the addition of the CFTR corrector lumacaftor [177]. Similarly, impaired phagocytosis and killing of *P. aeruginosa* by F508del homozygous human monocyte-derived macrophages were restored to levels seen in healthy control macrophages following culture with lumacaftor [178]. A recent study by Loske et al. revealed that ETI therapy markedly reduced the inflammatory phenotype of immune cells by reducing the activation of monocyte-derived/non-resident macrophages and neutrophils in the nasal mucosa of children with CF [166]. Treatment with ivacaftor restored phenotypic polarization and phagocytosis in monocyte-derived macrophages (MDMs) obtained from pwCF to levels similar to healthy donor MDMs [179]. Ivacaftor is also reported to correct the delayed neutrophil apoptosis, which is the primary contributor to non-resolving CF airway inflammation, in individuals with CFTR-G551D mutations [180]. Previous studies have also examined the effects of CFTR corrector therapy on inflammasome activation and found that monocytes from individuals treated with ETI demonstrated decreased expression of the P2 × 7 receptor, which promotes ATP-induced inflammasome activation, and were more resistant to inflammasome activation than monocytes from the same individuals before receiving ETI [181]; tezacaftor/ivacaftor treatment has been linked to similar changes in inflammasome activation in a study of PBMCs from patients with CF homozygous for F508del mutations [182]. Administration of lumacaftor/ivacaftor to pwCF homozygous for F508del mutations was also shown to increase the key metabolic checkpoint molecule PTEN levels in PBMCs [183], suggesting that highly effective CFTR modulator therapy will shift cellular metabolism in vivo to a less inflammatory state [170], therefore tuning down the characteristic hyperinflammatory disease state associated with CF-ABPA.

## 6. Effect of CFTR Modulators on ABPA Biomarkers and *Af*-Mediated Airway Infection

In this Section, we have outlined the impact of CFTR modulators on CF-ABPA particularly focusing on the ABPA diagnostic biomarker IgE and *Af* infection and colonization.

### 6.1. Impact of CFTR-Modulator Therapy on ABPA Biomarkers

In ABPA the patient’s total IgE is a diagnostic marker and often tracks with clinical exacerbation of disease. The effect of ETI therapy on total serum IgE, as a marker of the Th2 inflammatory response was examined in retrospective studies for adult patients with CF [31,184,185,186]. Mean total IgE decreased by approximately 20% [31] and 68% [184] following ETI therapy initiation. Mehta et al. investigated the correlation between post-ETI initiation change in IgE and lung function and found that the reduction in total IgE following ETI therapy did not significantly differ by initial disease severity. After starting ETI, patients with at least one positive fungal culture prior to ETI experienced a significant reduction in mean total IgE, whereas the patients with no history of fungal colonization did not [31]. In a study involving 198 pwCF in West Scotland, specific *Af* IgG and IgE-M3 significantly decreased following ETI therapy, likely contributing to improvements in FEV_1_ and body mass index [186]. Another similar study, examining *Aspergillus* IgG and IgE, 3 years before the initiation of ETI and at 3-month intervals for 12 months, found that *Aspergillus*-IgG and IgE significantly declined after 3 months of ETI initiation, which plateaued thereafter [185]. Thus, CFTR modulator therapy leads to a decreased allergic response, especially the *Af*-mediated allergic response. As ABPA is always associated with *Af*-induced allergic IgE-related reactions, administration of CFTR modulators might be a promising therapeutic approach for patients with CF-ABPA to control the exacerbation of allergic inflammation. 

### 6.2. Impact of CFTR Modulator Therapy on Af Infection and Colonization

Investigations have examined the immunologic and biologic mechanisms underlying the efficacy of CFTR modulators against *Af* [12]. ETI therapy was found to impede *Af* biomass growth, disrupt cell viability, and increase membrane permeability [187]. Interestingly, elexacaftor alone significantly reduced total *Aspergillus* biomass and the elexacaftor/ivacaftor combination was as effective as the triple-ETI combination, suggesting the importance of elexacaftor and ivacaftor in targeting *Af*. Notably, the study also found ETI modulates ion channels in *Aspergillus* biofilms, as ETI-induced permeability and biomass reduction were partially mitigated by a mammalian CFTR inhibitor GlyH-101, suggesting a direct biological mechanism for ETI’s antifungal activity in CF patients. Additionally, this modulation is associated with a decrease in inflammatory properties of *Af* with lower levels of inflammatory cytokine TNF-⍺ following ETI treatment, compared to controls without ETI [187]. 

The decrease in pulmonary exacerbation by CFTR modulator therapy is likely multifactorial, with one such factor being the reduction in bacterial and fungal colonization associated with modulator therapy. Several retrospective cohort studies have reported a CFTR modulator-mediated reduction in *Aspergillus* infections in pwCF. This reduction was first noted in ivacaftor monotherapy [188]. An annual analysis comparing ivacaftor-treated and untreated pwCF from the United States and the United Kingdom, performed by Bessonova et al., concluded that *Aspergillus* positive cultures were less prevalent among the ivacaftor-treated cohort (10.7% vs. 18.8% in the United States group: 10.3 vs. 20.2% in the United Kingdom group) compared to the comparators who had never received ivacaftor [188]. In ivacaftor-treated patients, the percentage of respiratory cultures positive for *Af* decreased from 30.4% to 18.7% in 12 months following initiation and maintained through 48 months [148]. This decrease could be attributed to ETI therapy, where the frequency of *Aspergillus*-positive sputum cultures was reduced with treatment initiation, which coincided with a significant decrease in anti-*Aspergillus* antibodies, as well as total IgE [189]. Concurrent reduction in *Aspergillus*-specific antibodies and IgE have suggested implications on the impact of modulator therapy on the prevalence of ABPA, which affects 8.9% of pwCF [72]. A large retrospective cohort study by Frost et al., spanning from the initiation of ivacaftor treatment in 2013 to its completion in 2016, compared 276 patients receiving treatment with ivacaftor with 5296 not receiving CFTR modulator therapy [190]. They observed a significant decrease in the prevalence of respiratory cultures positive for *Af*, from 12.0% to 4.7% (Adjusted Prevalence Ratio = 0.56). Conversely, untreated individuals showed stable *Af* culture positivity (16.8% to 16.9%) during the same period. 

ETI therapy has also demonstrated favorable effects on *Af* serology in adults with CF. Ex vivo analyses of a 21 CF patient study showed a significant decrease in antigen-specific CD154 (+) T cells against *Af* (−50.2%), as well as reductions in total serum IgG and IgE levels following ETI initiation [184]. The post-ETI therapy reduction in antigen load is thought to be the effect of improved mucociliary clearance after ETI initiation. These data are further supported by a significant decrease in *Af* sputum detection in the cohort and overall, the study suggests that CFTR restoration of B and T cells function enables a more effective immune response to *Af* [184]. Several other studies similarly found that CF modulators result in the clearance of *Aspergillus*. An interim analysis of a pilot study, examining 166 home-collected samples of 44 subjects with a known history of *Aspergillus*, found that sputum volume and quality were lower and less dense than previous sputa before ETI initiation [191]. Similarly, a retrospective multi-chart review, including 55 patients, found significant reductions (29.3% vs. 6.9%) in the prevalence of *Aspergillus* among sputum cultures [192]. The PROMISE study, which utilized droplet digital polymerase chain reaction (ddPCR) to measure *Af* in CF sputum more precisely, revealed rapid clearance of sputum *Af* following ETI treatment [193]. ETI therapy is thought to improve mucociliary clearance, leading to reduced fungal, and other bacteria-related burdens, thereby decreasing Th2 cytokine production, which further mitigates IgE-mediated allergic inflammation [31]. Another proposed mechanism suggests that CFTR modulators decrease *Af* prevalence by downregulating *Af*-induced reactive oxygen species (ROS) production among CF phagocytes without compromising their fungal killing ability [194]. Pretreatment with ivacaftor alone or in combination with lumacaftor significantly diminished *Af*-induced ROS in peripheral blood mononuclear cells (PBMCs) and polymorphonuclear leukocytes (PMNs), from patients homozygous for the F508-del mutation, with non-F508del and non-G551D mutations. This study also highlighted the higher sensitivity of CF PBMCs in potentiating the channel function by ivacaftor pretreatment than CF PMNs. However, contradicting other reports that indicate CFTR modulators augment bacterial killing (especially *P. aeruginosa*), this study reported that pretreatment with CFTR modulators was not associated with enhanced fungal killing [194]. Thus, CFTR modulators may have immunomodulatory benefits to prevent *Af* colonization as well as *Af*-induced inflammation in CF; however, the exact mechanism of action is yet to be elucidated.

## 7. Conclusions and Future Scope of Work

The widespread application of CFTR modulator therapy has provided the greatest impact on life expectancy and quality of life for pwCF. Various studies have already demonstrated objective improvements in patient lung function, as well as a decrease in rates of pulmonary exacerbation. The review of the pertinent literature presents compelling evidence of the promise held by CFTR modulatory therapy in the management of CF-ABPA. CFTR modulator therapy directly addresses the underlying pathophysiology of CF, which restores ionic balance in epithelial cells and improves mucociliary clearance. Restoration of CFTR activity has been associated with improved cellular function and decreased inflammatory response, which is implicated in the pathology of ABPA. Additionally, the use of CFTR modulators such as ETI exhibits promising effects against *Af* colonization and infection, as evidenced by various clinical and basic science studies, thereby eliminating the initial trigger for disease formation.

Overall, there are multiple plausible physiologic pathways through which CFTR modulators may address ABPA (Figure 3). While the effects of modulators on cellular pathways have been established, studying the effects of these medications in vivo proves more challenging due in part to the lack of standardized diagnostic criteria for ABPA, especially within pwCF. Ecological data from the CFF suggest that the prevalence of ABPA may be decreasing, coinciding with the expansion of modulator therapies. However, further research is warranted to elucidate the clinical impact of CFTR modulators, particularly due to limitations in sample size in several studies, and to comprehensively understand the alterations in the respiratory microbiome influencing *Af* dynamics in CF patients. We acknowledge that there are limitations to our current literature review. Due to the lack of large-scale randomized control trials evaluating the impact of CFTR modulator therapy on CF-ABPA disease outcome, a meta-analysis was beyond the scope of this current review. As a result, ours is a narrative review that rather than performing any statistical analysis, provides a narrative synthesis of the available literature, reaffirming the need for future clinical studies in this field. Nonetheless, with increasing patient populations eligible for CFTR modulators, understanding the intricacies of the interaction between these medications and ABPA is essential to further decreasing the lingering morbidity and mortality associated with CF.

## Figures and Tables

**Figure 1 jof-10-00656-f001:**
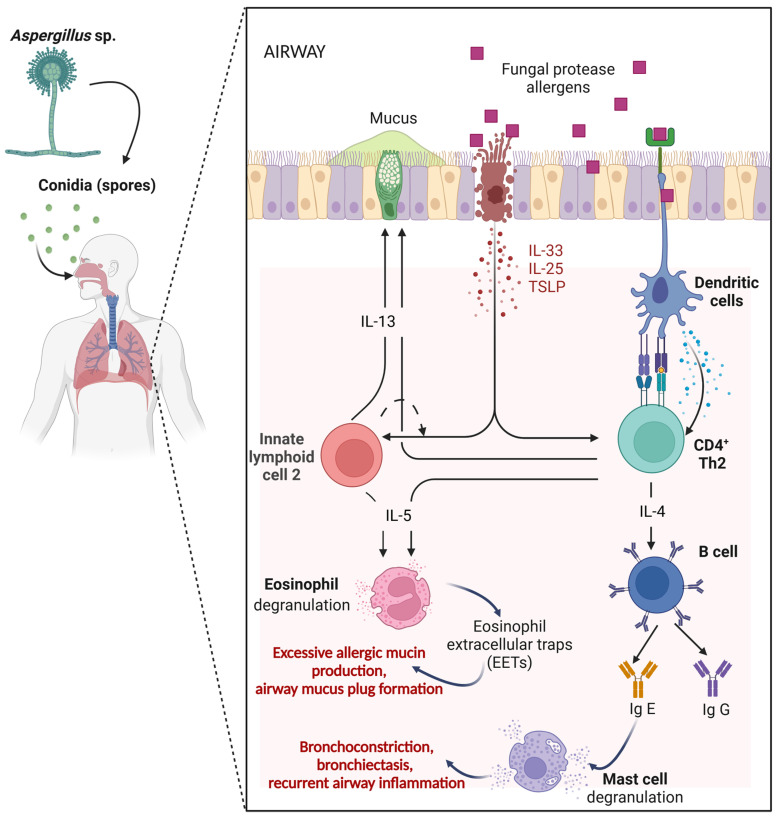
Pathophysiology of ABPA. Detailed framework involving the molecular and cellular elements of the local innate response to *Aspergillus* in driving a T-helper cell-2 (Th2) adaptive immunity in the airway exposed to fungal protease allergens. Figure created with BioRender.com.

**Figure 2 jof-10-00656-f002:**
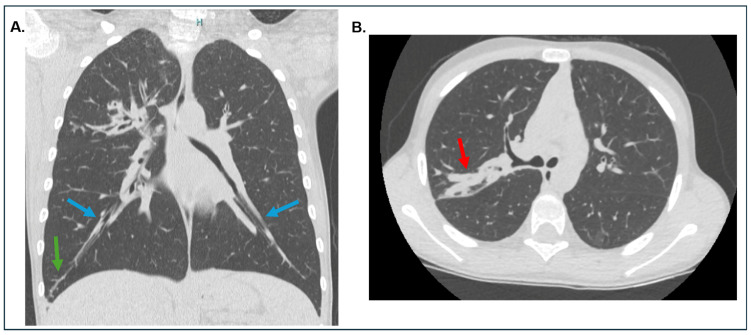
Computed tomography (CT) of chest. The CT chest of a pediatric patient with cystic fibrosis complicated by allergic bronchopulmonary aspergillosis showing: (**A**) bronchiectasis with the hallmark findings of dilated bronchi with bronchial wall thickening (blue arrows) and tree-in-bud pattern opacities (green arrow) indicating terminal airway mucus impaction and peribronchiolar inflammation; and (**B**) “finger in glove sign” (red arrow) characterized by tubular opacification resulting from mucus plugging of branching dilated bronchioles.

**Figure 3 jof-10-00656-f003:**
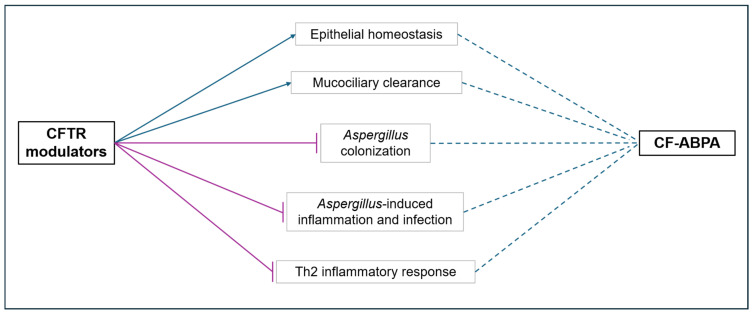
Schematic diagram of possible mode of actions for CFTR modulators on CF-ABPA. Possible positive impacts of CFTR modulators are represented by blue arrows and possible negative impacts are represented in purple blunt head arrows, which implicates the indirect but favorable effects (represented by blue dotted line) of CFTR modulators on CF-ABPA.

**Table 1 jof-10-00656-t001:** Current diagnostic criteria for ABPA determined by ISHAM.

Criteria	Essential Criteria	At Least One Criterion Required
Criteria 1	1. Serum *Af*-specific IgE > 0.5 kUA/L	1. Serum *Af*-specific IgG > 27 mgA/L
	2. Serum total IgE > 500 IU/mL	2. Bronchiectasis on CT chest scan
Criteria 2	1. Serum *Af*-specific IgE > 0.5 kUA/L	1. TEC > 500 cells/µL
	2. Serum total IgE > 500 IU/mL	2. Bronchiectasis on CT chest scan
Criteria 3	1. Serum *Af*-specific IgE > 0.35 kUA/L 2. Serum total > 500 IU/mL 3. Bronchiectasis on CT chest scan	
Criteria 4	1. Type 1 *Aspergillus* skin test positive 2. Serum total IgE > 500 IU/mL 3. Bronchiectasis on CT chest	
Criteria 5	1. Serum *Af*-specific IgE > 0.35 kUA/L	1. Serum *Af*-specific IgG > 27 mgA/L
	2. Serum total IgE > 500 IU/mL	2. Bronchiectasis on CT chest
Criteria 6	1. Serum *Af*-specific IgE > 0.35 kUA/L	1. Bronchiectasis on CT chest
	2. Serum total IgE > 500 IU/mL	2. TEC > 500 cells/µL
Criteria 7	1. Serum *Af*-specific IgE > 0.35 kUA/L	1. Serum *Af*-specific IgG > 27 mgA/L
	2. Serum total IgE > 500 IU/mL	2. TEC > 500 cells µL

ABPA, allergic bronchopulmonary aspergillosis; *Af*, *Aspergillus fumigatus*; CT, computerized tomography; IgG, immunoglobulin G; IgE, immunoglobulin E; ISHAM, International Society for Human and Animal Mycology; TEC, total eosinophil count (peripheral).

**Table 2 jof-10-00656-t002:** Medications being used to treat ABPA.

Class	Drugs	Mechanism of Action	Adverse Effects		References
*Corticosteroid*					
Glucocorticoid	Prednisone Prednisolone Methylprednis-olone	Inhibition of phospholipase A_2_, chemotaxis, and cytokine production, decreasing eosinophilia	Fluid retention, hyperglycemia, hypertension, increased appetite, weight gain, acne, osteonecrosis, osteoporosis, adrenal insufficiency, glaucoma, pancreatitis		[20,111,121,122,123,124]
*Antifungals*					
Triazole	Itraconazole	Inhibition of a cytochrome P450 14-α-sterol demethylase enzyme	Hepatotoxicity, hypokalemia, nausea, emesis, diarrhea	Myositis, peripheral neuropathy, adrenal insufficiency, pancreatitis, peripheral edema, QTc prolongation	[111,125,126,127,128,129,130,131]
	Voriconazole			Periostitis, myositis, peripheral neuropathy, adrenal insufficiency, hyponatremia, pancreatitis, squamous cell carcinoma, QTc prolongation, visual impairments: blurred vision, photophobia, color blindness; cardiotoxicity	
	Isavuconazole			Peripheral edema, QTc shortening	
*Biologics—monoclonal antibodies (mAbs)*					
Humanized mAbs	Omalizumab	Selective binding to IgE, inhibiting activity resulting in downregulation of FcεRI on basophil and mast cells	Injection-site reaction, headache, anaphylaxis	Cancer, helminth infection, arthralgia, viral infection, upper respiratory infection, pharyngitis	[41,111,122,132,133,134,135,136,137,138,139,140,141,142,143]
	Mepolizumab	Selective binding to IL5, inhibiting interaction with IL5R, decreasing activity of eosinophils		Herpes zoster, arthralgia, upper respiratory infection, viral infection	
	Benralizumab	Selective binding to IL5Rα on basophils and eosinophils leading to apoptosis via antibody-dependent cell-mediated cytotoxicity		Pyrexia, pharyngitis	
Human-derived mAbs	Dupilumab	Selective binding to IL4Rα, inhibiting signaling of IL-4 and IL-13 thereby reducing cytokines, chemokines, and IgE		Conjunctivitis, keratitis, herpes simplex infection, arthralgia, gastritis	

FcεRI, high-affinity IgE receptor; IgE, immunoglobulin E; IL-4, interleukin 4; IL4Rα, interleukin 4 receptor alpha; IL-5, interleukin 5; IL5Rα, interleukin 5 receptor alpha; IL-13, interleukin 13; mAb, monoclonal antibody.

**Table 3 jof-10-00656-t003:** Current CFTR modulator therapies.

Modulator Class	Mechanism of Action	Drugs	References
Potentiator	enhances the probability of CFTR channels opening (gating)	Ivacaftor	[15,16,144]
Corrector	binds to misfolded CFTR proteins, assisting in the folding process	Elexacaftor * Lumacaftor Tezacaftor	[13,17,144]
Amplifier	increases the synthesis of CFTR protein	Nesolicaftor	[145]
Read through	remediate stop signals during CFTR protein synthesis	Ataluren	[146]
Stabilizer	increases stability of misfolded CFTR protein on cell surfaces	None available	[147]

* Elexacaftor is a corrector with additional potentiator activity. FDA-approved drug information is available at https://dailymed.nlm.nih.gov/dailymed (accessed on 30 March 2024).

## Data Availability

No new data were created for this study.

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
