# Peer review of "Allergic Bronchopulmonary Aspergillosis (ABPA) in the Era of Cystic Fibrosis Transmembrane Conductance Regulator (CFTR) Modulators"

_jof, 2024, doi:10.3390/jof10090656_

Round 1
Reviewer 1 Report
The submitted by Chaterjee P et al manuscript is well designed and conducted. The authors could shorten the introduction.
No additional comments
Author Response
The authors could shorten the introduction section.
Response: Thank you for your encouraging comments. Based on your suggestion, we have revised the background section and shortened the introduction portion.
Reviewer 2 Report
Reviewer’s report on Allergic Broncho-pulmonary aspergillosis (ABPA) in the era of cystic fibrosis transmembrane conductance regulator (CFTR) modulators
The work of Chatterjee and colleagues represents a very fine work. Authors have performed an outstanding evaluation of a research matter that has not been completely elucidated to date. References are well placed and paced through the text. Although the complexity of the arguments treated and the relatively paucity of works published regarding this specific arguments authors managed to deliver a detailed and exhaustive literature research. This review provides new insights on the clinical intersection between Allergic bronchopulmonary aspergillosis in Cystic fibrosis patients in the era of newly available treatments such as transmembrane conductance regulator (CFTR) modulator therapy, in particular elexacaftor/tezacaftor/ivacaftor.
I have only minor comments regarding this review to be appointed by authors:
Abstract.
Line 16. All species name must be written in italics.
1.2 Allergic Bronchopulmonary Aspergillosis: Despite being extensively and meticulously described; I would suggest adding an explanatory figure of the pathogenesis of ABPA.
1.3 CF-ABPA: Line203. Please add a reference to this sentence.
3 Treatment for CF-ABPA: Line 290. Table 2. I would recommend to add proper reference to each medication presented.
4. CFTR Modulator Therapy. Summing up the most relevant information of these chapter in one figure would help the reader following the detailed and dense-of-concept description of the molecular effect of CFTR therapy.
Author Response
We sincerely thank the reviewer for their insightful comments and constructive feedback on our manuscript. Their suggestions have significantly improved the quality and clarity of our work.
Line 16. All species name must be written in italics.
Response: We have made the suggested change in line 16.
1.2 Allergic Bronchopulmonary Aspergillosis: Despite being extensively and meticulously described; I would suggest adding an explanatory figure of the pathogenesis of ABPA.
Response: Based on the reviewer’s suggestion, we have added a figure (Fig 1 in the revised manuscript) on the pathogenesis of ABPA.
1.3 CF-ABPA: Line203. Please add a reference to this sentence.
Response: We have added references to this line accordingly.
3 Treatment for CF-ABPA: Line 290. Table 2. I would recommend to add proper reference to each medication presented.
Response: We have added references in the Table 2 corresponding to each class of medications.
- CFTR Modulator Therapy. Summing up the most relevant information of these chapter in one figure would help the reader following the detailed and dense-of-concept description of the molecular effect of CFTR therapy.
Response: We thank the reviewer for this excellent suggestion. While we recognize that summing up all the information in a figure might be a comprehensive way to report the molecular effects for CFTR modulators, we feel this figure will be redundant in this section as Table 2 of this review paper already summarizes the classification and mode of action of the known CFTR modulators in a succinct way.
Reviewer 3 Report
The manuscript presents a literature review to examine the impact of ETI therapy, used in FC, on ABPA, especially on allergic immune response pathways and Af infection.
The topic is very interesting, but I think that a narrative review does not greatly benefit the presentation and exposition of the topic. A meta-analysis to evaluate the impact of the therapy would be advisable.
Abstract
No results or conclusions are presented in the abstract.
The manuscript is too long, the historical information is interesting, but I think it contributes little to the main topic.
There is repetitive information between Table 2 and the information in points 3.1 to 3.3, particularly regarding adverse effects. It is necessary for authors to review the information because what is presented in the tables should not be in the text and vice versa.
In section 1. Background I suggest that the relationship between cystic fibrosis and ABPA be addressed. Likewise, in 2. Diagnosis of ABPA it would be advisable to focus on the diagnosis of ABPA in people with cystic fibrosis rather than focusing on historical details.
The manuscript does not mention the limitations of the work. The main limitation I see is that this is a narrative review and not a systematic review.
Conclusions
The authors mention: "It is reassuring that our review demonstrated few reports of the development of ABPA in patients treated with CFTR modulators..." However, as this is a narrative review, I consider it inappropriate to state this.
Table 2. and Table 3 A column with the references is missing
References
It presents very old references, which are important but I think it should be something more updated and not so historical.
Table 2. and Table 3 A column with the references is missing
Author Response
We sincerely thank the reviewer for their insightful comments and constructive feedback on our manuscript. Their suggestions have significantly improved the quality and clarity of our work.
No results or conclusions are presented in the abstract.
Response: According to the reviewer’s suggestion, we have incorporated results and conclusions in the abstract, line 30-36.
The manuscript is too long, the historical information is interesting, but I think it contributes little to the main topic.
Response: Based on the reviewer’s suggestion, we have thoroughly revised the entire manuscript, especially the background section (page 2) and the diagnosis section (page 7-8) to eliminate the historical details and shorten the manuscript.
There is repetitive information between Table 2 and the information in points 3.1 to 3.3, particularly regarding adverse effects. It is necessary for authors to review the information because what is presented in the tables should not be in the text and vice versa.
Response: Based on the reviewer’s suggestion, we have removed the repetitive information on adverse effects from sections 3.1, 3.2 and 3.3.
In section 1. Background I suggest that the relationship between cystic fibrosis and ABPA be addressed. Likewise, in 2. Diagnosis of ABPA it would be advisable to focus on the diagnosis of ABPA in people with cystic fibrosis rather than focusing on historical details.
Response: The relationship between cystic fibrosis (CF) and ABPA has been extensively discussed in the section 1.2 Allergic Bronchopulmonary Aspergillosis (pages 3-5), which highlights the risk factors responsible for predisposing CF patients to ABPA, and section 1.3: CF-ABPA (pages 6-7), which highlights the epidemiologic incidences of ABPA in CF patients.
We have revised section 2 to address the reviewer’s comments and incorporated detailed discussion on the minimal diagnostic criteria for the diagnosis of ABPA in people with CF patients (lines 277-293). We have also removed historical details on ABPA disease staging according to the reviewer’s suggestion.
The manuscript does not mention the limitations of the work. The main limitation I see is that this is a narrative review and not a systematic review.
Response: We have revised the conclusion section and added limitations for our study according to the reviewer’s suggestion (page 18, line 691-697).
The authors mention: "It is reassuring that our review demonstrated few reports of the development of ABPA in patients treated with CFTR modulators..." However, as this is a narrative review, I consider it inappropriate to state this.
Response: We have revised this section accordingly.
Table 2. and Table 3 A column with the references is missing
Response: We have added references in both Table 2 and Table 3, accordingly.
References
It presents very old references, which are important but I think it should be something more updated and not so historical.
Response: Based on the reviewer’s suggestions, we have updated the references, by incorporating some newer references and removing some of the historical references.
Round 2
Reviewer 3 Report
Thanks for taking the suggestions.
I have no more suggestions.
Author Response
We appreciate the encouraging feedback from the reviewer and sincerely thank them for reviewing our revised manuscript.